# Single-Session Impact of High-Power Laser with Moses Technology for Lower Pole Stones in Retrograde Intrarenal Surgery: Retrospective Study

**DOI:** 10.3390/jcm12010301

**Published:** 2022-12-30

**Authors:** Takaaki Inoue, Shuzo Hamamoto, Shinsuke Okada, Fukashi Yamamichi, Masaichiro Fujita, Koki Tominaga, Yasumasa Tobe, Masato Fujisawa

**Affiliations:** 1Department of Urology and Stone Center, Hara Genitourinary Hospital, 5-7-17 Kitanagasadori, Kobe 650-0012, Japan; 2Division of Urology, Department of Surgery Related, Kobe University Graduate School of Medicine, Kobe 650-0017, Japan; 3Department of Urology, Medical School, Nagoya City University Graduate School of Medical Sciences, Nagoya 467-8601, Japan; 4Department of Urology, Gyotoku General Hospital, Ichikawa 272-0103, Japan

**Keywords:** lower pole stone, flexible ureteroscopy, retrograde intrarenal surgery, Moses technology

## Abstract

**Background:** This study aimed to evaluate the efficacy of a high-power holmium laser with Moses technology (MT) for the treatment of lower pole stones during retrograde intrarenal surgery (RIRS). **Methods:** Herein, 305 patients with lower pole stones who underwent RIRS using a high-power holmium laser with MT were retrospectively classified into the stone-free (SF) and non-SF groups. We measured the stone burden, stone volume, stone hardness, pre- or post-operative stent placement, infundibulopelvic angle (IPA), infundibular width (IW), infundibular length (IL), and calyceal pelvic height in terms of pelvicalyceal anatomy using retrograde pyelograms and evaluated the predictive factors of postoperative SF. **Results:** A total of 173 (56.7%) and 229 (75.1%) patients achieved a SF status on postoperative day one and at one month, respectively. Operation time in the SF group was shorter than that in the non-SF group (51.0 vs. 74.5 min). There were no significant differences in postoperative complications between the SF and non-SF groups. Significantly predictive risk factors in postoperative SF included total stone volume (odds ratio (OR), 1.056; 95% CI, 1.015–1.099; *p* = 0.007), IPA (OR, 0.970; 95% CI, 0.956–0.993; *p* = 0.009), and IW (OR, 0.295; 95% CI, 0.121–0.718; *p* = 0.007). The cut-off values of stone volume, IPA, and IW were 515.2 mm^3^, 46.8°, and 7.75 mm, respectively. **Conclusions:** A high-power holmium laser with MT in lower pole stones is a valuable option for positive outcomes and patient’s safety. Larger stone volume, acute IPA, and narrow IW were negative predictors related to postoperative SF status.

## 1. Introduction

During the development of our endourological innovation, the technical skill of a stone treatment during retrograde intrarenal surgery (RIRS) rapidly changed. A holmium YAG laser is already one of the standard tools used to disintegrate urinary tract stones. Currently, high-power holmium YAG lasers, which equip pulse frequencies higher than 100 Hz with Moses technology (MT), and thulium fiber lasers are available in clinical practice. These laser innovations have made it possible to create much smaller stone dusts compared with traditionally creating and removing stone fragments [1]. Therefore, the trends in stone lithotripsy have gradually changed toward producing stone dust, with as much spontaneous passage expected after operation as possible. However, even though using novel laser systems results in much smaller target stone dusts during RIRS, not every case can expect spontaneous passage completely and becomes stone-free (SF) because of multifactorial influences, such as stone size, stone hardness, stone location, pelvicalyceal anatomy, etc. [2]. Some investigators reported that the SF rates of MT and thulium fiber lasers in kidney stones were 52.3–79.3% and 85.7–92.5%, respectively [3,4,5]. However, there have been insufficient clinical data on treatment outcome when using a high-power holmium laser with MT for lower pole stones. Therefore, we evaluated the efficacy and safety of a high-power laser with MT for lower pole stones and analyzed the predictive factors of postoperative SF in the present study.

## 2. Materials and Methods

### 2.1. Study Design and Patients

Our institutional review board approved this retrospective study (approval No. 20216102). We acquired patient’s informed consent with an opt-out letter in our hospital, and 305 consecutive patients who underwent RIRS using flexible ureteroscope (fURS) and high-power holmium laser treatment with MT for lower pole stones between April 2019 and January 2022 at our institution were enrolled. Patients with congenital anomalies and the simultaneous presence of ureteral, middle, or upper pole stones were excluded. Our primary endpoint was the efficacy and safety of a high-power holmium laser with MT for lower pole stones. Our secondary endpoint was the analysis of predictive risk factors to archive SF for lower pole stones after RIRS. 

We classified the patients into the SF and non-SF (n-SF) groups. The criteria for postoperative SF status were defined as no residual stone in low dose computed tomography (LDCT) at one month postoperatively.

### 2.2. Data Collection and Measurement of Pelvicalyceal Anatomy

Data pertaining to the following aspects were collected from the patient’s medical records and imaging study reports: age; gender; body mass index (BMI); stone laterality; preoperative stent placement; and stone demographics, such as stone location, total maximum stone size, stone volume, and stone density. Additionally, surgical outcome, including SF status on postoperative day one (POD 1) using ultrasonography for the Kidneys, ureters, and bladder (KUB) and one-month postoperatively using LDCT; ureteral access sheath (UAS); UAS size; postoperative stenting; operation time; total laser energy; and intra- and post-operative complications were collected. Intraoperative ureteral injury grade was classified using the Traxer and Thomas criteria, as follows [6].

The total stone size was calculated by adding the length of the longest axis of each stone from the CT image. The total stone volume was also calculated by adding each stone volume, which was calculated using the ellipsoid formula (0.167 × π × Length × Width × Height), and the stone density in Hounsfield Unit (HU) was the maximum point value found within the circle enclosing the whole stone. 

The pelvicalyceal anatomy in thte collecting system was accessed in all patients by a retrograde pyelogram with contrast material under fluoroscopy while undergoing RIRS. We measured the IPA, IW, and IL, as previously described by Elbahnasy et al. [7], and the calyceal pelvic height (CPH), as defined by Tuckey et al., with a retrograde pyelogram [8,9] (Figure 1A,B). Two urologists (T.I. and F.Y.) took these measurements.

### 2.3. Surgical Steps and Postoperative Management 

All surgeries were carried out under general anesthesia by two stone experts in our high-volume center. The surgical steps in RIRS were as follows: First, 6-Fr semi-rigid ureteroscope over a guidewire (Sensor™, Boston Scientific, Marlborough, MA, USA) was routinely inserted into the diseased side ureter to observe the ureteral lumen size. Then, the surgeon decided the suitable UAS diameter and length according to routine endoscopic ureteral observation. After placing the UAS in the ureter, the fURS (URF-P7^®^; Olympus, Tokyo, Japan, or FlexX2s; KARL STORZ, Germany) was inserted through the UAS. If the UAS could not be inserted through the ureter, the fURS was inserted directly into the ureter over the guidewire. A 120 W holmium-yttrium aluminum garnet laser (Pulse 120H, Lumenis, Yokneam, Israel) with a 200 μm end-firing laser fiber (MOSES^TM^ 200 D/F/L Smooth tip fiber, Luimenis, Israel) was utilized to disintegrate the stone. First, we used our strategy of laser setting, 6–10 Hz and 0.6–1.2 J in MOSES Contact mode, to disintegrate the stone as fragments, and then, 80 Hz with 0.5 J from MOSES distance to Contact mode was finally used to disintegrate the stone into fine dust like pop-dusting into the renal calix. Larger quarried fragments were removed using a 1.5-Fr nitinol basket (N-circle^®^, Cook Medical, Bloomington, IN, USA). However, a lot of stone dust, which could not be extracted, remained in the renal collecting system. At the end of the surgery, the contrast material was injected through the working channel of fURS at the lower pole calyx to draw the pelvicalyceal anatomy until the whole upper, middle, and lower calyces were enhanced under fluoroscopy. Additionally, then, digital images of fluoroscope were captured, and the factors of pelvicalyceal anatomy, such as IPA, IW, IL, and CPH were measured after operation. Finally, we checked if ureteral injury occurred under direct ureteroscopic vision and then decided the need for postoperative stenting. If there was no any ureteral injury along the whole ureteral length and no stone fragments of <2 mm except the stone dust, we proactively archived less ureteral stent after the RIRS. The ureteral stents of almost all the patients with postoperative stenting were removed in our outpatient clinic on postoperative week 1–2. 

### 2.4. Statistical Analysis

IBM SPSS Statistics version 26 was used to analyze all clinical data. Continuous and categorial variables are described as medians with interquartile ranges and numbers with percentages, respectively. Furthermore, the chi-square test and Mann–Whitney U test were used to conduct the univariate analysis and logistic regression for multivariable analysis to compare the groups. Receiver operating characteristic curves with the Youden index were utilized to measure the appropriate cut-off value of continuous variables for SF status. A two-sided *p*-value of <0.05 was considered statistically significant.

## 3. Results

Of the 305 patients, 173 (56.7%) and 229 (75.1%) patients achieved SF on POD one and at one month, respectively. Although patient’s characteristics were not significantly different between the SF and non-SF groups, median total stone size, total stone volume, and stone density were significantly different in 14.0 and 24.5 mm (*p* < 0.001), 325.5 and 767.5 mm^3^ (*p* < 0.001), and 1045 and 1259 HU (*p* = 0.005), respectively. In terms of renal pelvicalyceal anatomy, IPA, IW, IL, and CPH between the SF and non-SF groups were 58.0° and 47.4° (*p* = 0.001), 10.2 and 8.8 mm (*p* = 0.005), 34.6 and 38.6 mm (*p* = 0.077), and 28.1 and 30.9 mm (*p* = 0.009), respectively (Table 1). 

Operation time in the SF group was significantly shorter than in the non-SF group (51.0 vs. 74.5 min, *p* < 0.001). The rate of postoperative complication was 6.2%. Although there were no significant differences in postoperative complications between the SF and non-SF groups, the rate of ureteral injury in intra-operative complications was higher in the non-SF group (1.7 vs. 10.5%, *p* = 0.001) (Table 2). Significant predictive risk factors in postoperative SF were total stone volume (odds ratio (OR), 1.056; 95% CI, 1.015–1.099; *p* = 0.007), IPA (OR, 0.970; 95% CI, 0.956–0.993; *p* = 0.009), and IW (OR, 0.295; 95% CI, 0.121–0.718; *p* = 0.007) (Table 3). The cut-off values of stone volume, IPA, and IW were 515.2 mm^3^, 46.8°, and 7.75 mm, respectively (Figure 2A,B).

## 4. Discussion

In this study, our data showed the SF rate at 75.1% for lower pole stones with a median size of 14.0 mm when using a high-power holmium laser with MT. The SF rate was generally 75.3–82.1% when using regular laser mode without MT for 10–20 mm lower pole stones [10,11]. In a study by Margaret et al., they found that the SF rate for patients with <20 mm kidney stones between a high-power holmium laser with MT and the regular mode without MT were 52.3% and 65.3%, respectively (*p* = 0.143). They concluded that there may be technical benefits to the MT not captured in their analysis, except SF status [4]. In contrast, Amelia et al. reported that compared with a low-power holmium laser (20 W), a high-power holmium laser with MT in lower pole stones of medium size (9.6–13.8 mm) had a higher SF rate (91.6 vs. 96.5%, *p* = 0.13; no significant difference) and shorter operating time (*p* < 0.001) [12]. Therefore, although the efficacy of a high-power holmium laser with MT in lower pole stones with medium size might not be higher than one of a regular holmium laser without MT with regard to SF status, there might be a great advantage that operation time is definitely faster than that with a low-power holmium laser.

Our complication rate was also lower in 6.2% of patients. According to some literatures, the postoperative complication rate after using a high-power holmium laser with MT for lower pole stones was slightly lower in 4.3–5.2% of patients than the previously reported rate of 7.4–9.7% because of the shorter operative time [12,13,14].

We analyzed the predictive risk factors of postoperative SF when using a high-power laser with MT in lower pole stones. There are some measurement methods to evaluate stone burden, such as stone diameter, stone surface area, and stone volume. To as attain a SF status using flexible ureteroscopy, Ito et al. reported that stone volume was a more predictive stone burden than cumulative stone diameter [15]. Diamand et al. also found that volumetric measurement had a better ability to predict SF status before flexible ureteroscopy, and tridimensional volume is more accurate, as well [16]. In the present study, stone volume was one of critical factors to archive SF during retrograde intrarenal surgery in even lower pole stones. Furthermore, our cut-off value of stone volume to archive SF status was 515.2 mm^3^. Panthier et al. reported the evaluation of a free three-dimensional software for kidney stones before flexible ureteroscopy. They archived a 81% SF rate for a stone volume of 479 mm^3^ [17].

Additionally, IPA and IW in terms of pelvicalyceal anatomy were significant predictive factors to archive a SF status of lower pole stones in this study. Resorlu et al. first investigated >45° IPA and <15 mm stone size as positive risk factors to predict achieving a SF state on postoperative RIRS for lower pole stones using a preoperative intravenous urogram [18]. Jessen et al. and Inoue et al. also found <30° IPA as a negative risk factor to predict stone clearance on postoperative RIRS for lower pole stones using a preoperative intravenous urogram [2,9]. Our study found that an IPA of >46.8° was a significantly positive factor in attaining SF. Additionally, Karim et al. reported and reviewed that IW for successful and unsuccessful procedures after RIRS ranged between 6–9 mm and 5.5–7 mm, respectively [19]. Our study also found that an IW of >7.75 mm was a significantly positive factor in postoperative RIRS as well. 

From previous reports of regular holmium laser without MT, stone diameter, stone volume, stone hardness, and the anatomy of the renal collecting system were found to be predictive factors related to SF status after RIRS. Our study also showed that almost similar factors, such as stone volume, IPA, and IW, influenced a successful procedure, even though we used a high-power holmium laser with MT for lower pole stones with a medium size. Stone volume, accessibility to lower pole with fURS, and pelvicalyceal anatomy to facilitate spontaneous stone passage might be quite critical factors related to SF status. Therefore, not only single-use ureteroscopy to gain access into complicated lower poles but also novel devices with the functionality to excrete as much tiny stone dusts as possible such as a vacuum or absorbing device will be required to improve the postoperative stone-free status in lower pole calices, as well as an additional percutaneous approach.

In this study, the postoperative stenting rate in the SF group was lower than that in the non-SF group because of less ureteral injury, the smaller initial stone size, and the shorter operation time. In our previous study, we had many cases with less pain and comparable complication rates when not placing a ureteral stent on postoperative RIRS compared to placing a ureteral stent. Therefore, we proactively performed a stentless ureteral procedure for the selected patients in this study [20].

This study has some limitations. First, we removed the relatively larger stone fragments using a stone basket. Therefore, the outcome of this study does not show the SF status when performing whole stone dusting with MT from the beginning to the end of RIRS. Second, we defined SF status as no residual stone in LDCT at one month postoperatively. However, <2 mm or <4 mm residual stones in KUB with US at 1 and 3 months postoperatively were used as the definition of SF status in many previous studies. Therefore, although our definition might be strict compared to other definitions. Third, this is a retrospective study in a single center, which would have introduced inherent bias. Nevertheless, to the best of our knowledge, this is the first report to evaluate the predictors of SF status when using a high-power holmium laser with MT for lower pole stones. We clarified the importance of three predictive factors to attain a SF status in our study. 

## 5. Conclusions 

A high-power holmium laser with MT in lower pole stones is a valuable option in achieving a SF status and patient safety. Larger stone volume, acute IPA, and narrow IW were negative predictors related to postoperative SF status.

## Figures and Tables

**Figure 1 jcm-12-00301-f001:**
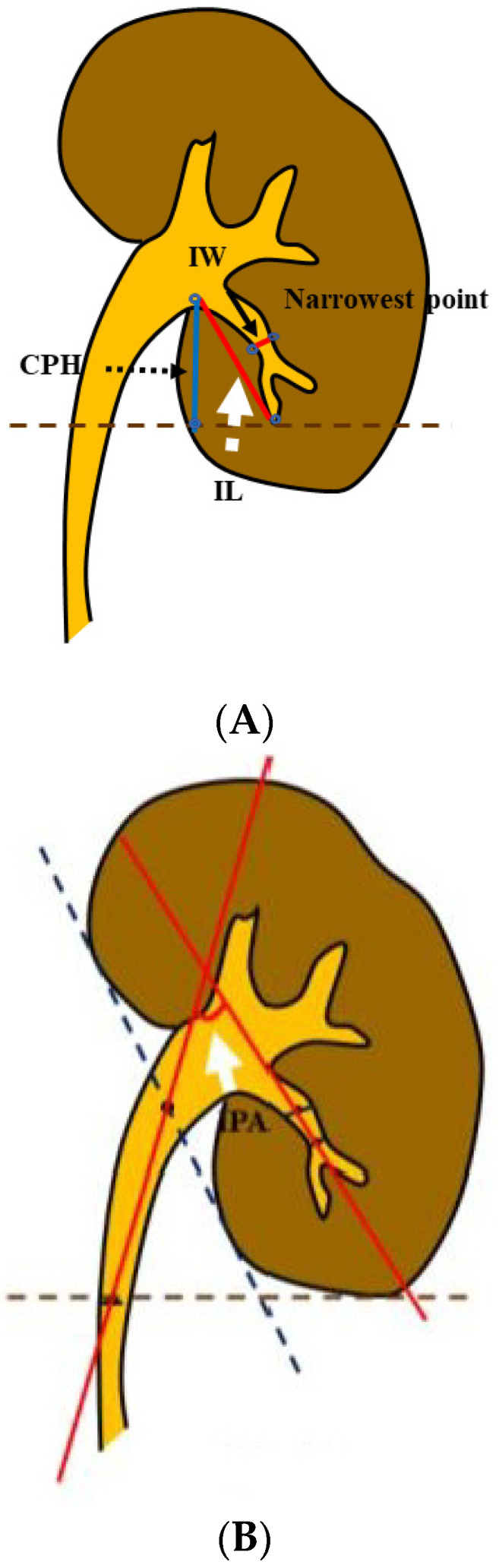
Spatial anatomical measurement of the renal collecting system. (**A**) The white dotted arrow indicates the infundibular length (IL); the slender black arrow indicates the infundibular width (IW); and the black dotted arrow indicates the calyceal pelvic height (CPH). (**B**) The white arrow indicates the infundibulopelvic angle (IPA).

**Figure 2 jcm-12-00301-f002:**
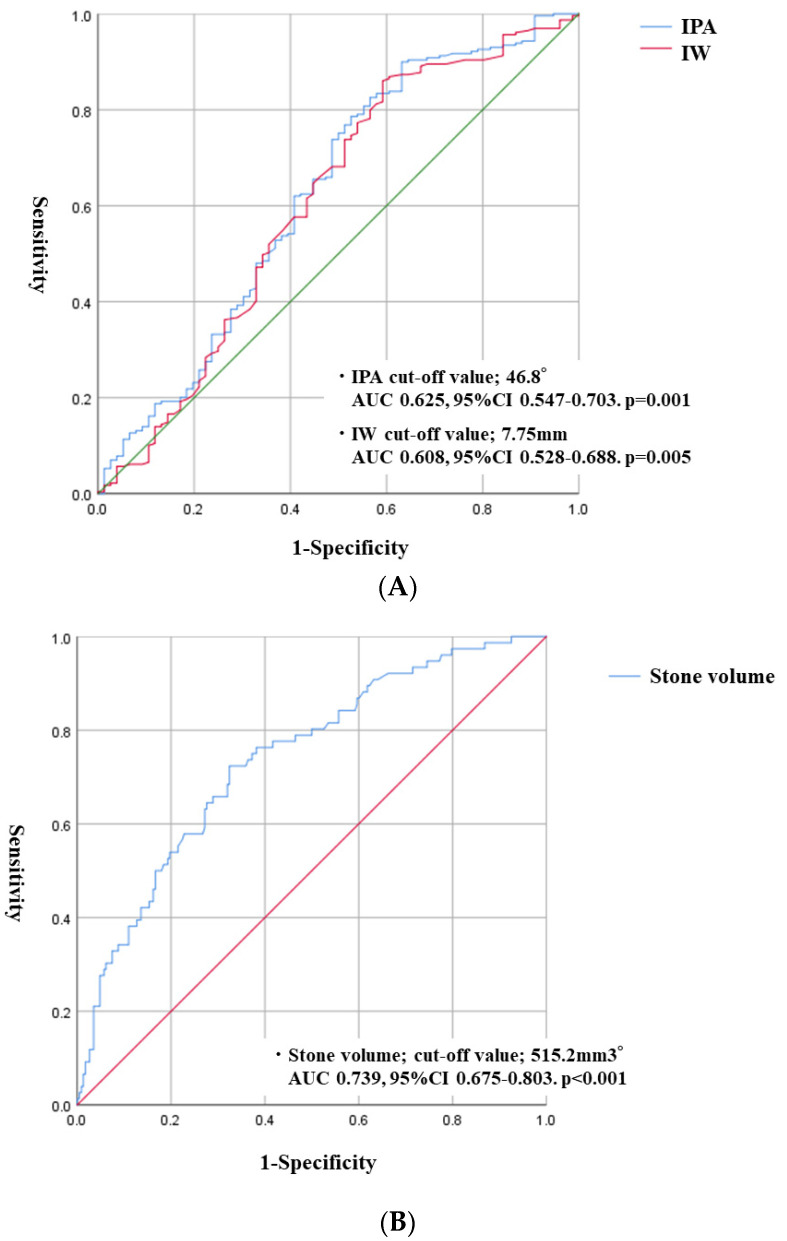
Receiver operating characteristic (ROC) curves used to calculate the optimal cut-off value of continuous variables for accessible lower pole calyces. (**A**) ROC curves indicate the IPA and IW cut-off value. (**B**) ROC curve indicates the cut-off value of stone volume.

**Table 1 jcm-12-00301-t001:** Patients, stone demographics, and pelvicalyceal anatomy in stone-free status.

<Patients *n* = 305>	Stone-Free Group (*n* = 229)	Non-Stone-Free Group (*n* = 76)	Univariate Analysis (*p*-Value) *
Age (yr, median, IQR)	59.0 (51.0–68.5)	59.5 (51.0–69.3)	0.880
Gender (No, %)	M; 146 (63.7) F; 83 (36.2)	M; 49 (64.4) F; 27 (35.5)	0.910
BMI (kg/m^2^, median, IQR)	23.6 (21.3–26.0)	23.6 (21.3–25.5)	0.934
Stone laterality (no, %)	Rt; 108 (47.1) Lt; 121 (52.8)	Rt; 32 (42.1) Lt; 44 (57.8)	0.443
Pre-operative stent placement (no, %)	179 (78.1)	55 (72.3)	0.300
<Stone>			
Stone location (No, %)			
Lower pole	229 (100)	76 (100)	-
Number of stone Simple Multiple	187 (81.6) 42 (18.3)	58 (76.3) 18 (23.6)	0.309
Total stone size (mm, median, IQR)	14.0 (10.0–21.0)	24.5 (15.0–32.0)	<0.001
Total stone volume (mm^3^, median, IQR)	325.5 (156.6–636.2)	767.5 (439.7–1311.5)	<0.001
CT-value (HU, median, IQR)	1045 (704.0–1385.0)	1259 (972.0–1480.0)	0.005
Stone composition (no, %)			0.633
CaOx/CaP UA Cystine Struvite Others	199 (86.8) 11 (4.8) 8 (3.4) 8 (3.4) 3 (1.3)	72 (94.7) 2 (2.6) 2 (2.6) 0 (0) 0 (0)	
<Renal pelvicaliceal anatomy>			
IPA (degree, median, IQR)	58.0 (47.4–71.1)	47.4 (35.2–65.4)	0.001
IW (mm, median, IQR)	10.2 (8.0–13.5)	8.8 (6.3–12.7)	0.005
IL (mm, median, IQR)	34.6 (29.4–41.0)	38.6 (30.1–42.7)	0.077
CPH (mm, median, IQR)	28.1 (21.3–33.6)	30.9 (23.6–39.0)	0.009

* Mann–Whitney U test and chi-squared test. BMI; Body Mass Index, HU; Hounsfield Unit, IQR; interquartile range, IPA; infundibulopelvic angle, IW; infundibular width, IL; infundibular length, CPH; caliceal pelvic height.

**Table 2 jcm-12-00301-t002:** Surgical outcome and post-operative complications in stone-free status.

	Stone-Free Group (*n* = 229)	Non-Stone-Free Group (*n* = 76)	Univariate Analysis (*p*-Value) *
<Surgical outcome>			
Placement of ureteral access sheath (no, %)	226 (98.6)	75 (98.6)	0.507
Size of ureteral access sheath (no, %) 9.5/11.5 Fr 10–12 Fr 10.7–12.7 Fr 11/13Fr No use of ureteral sheath	45 (19.6) 171 (74.6) 1 (0.4) 9 (3.9) 3 (1.3)	11 (14.4) 57 (75.0) 0 (0.0) 7 (9.2) 1 (1.3)	0.504
Postoperative stent placement (no, %)	151 (65.9)	66 (86.8)	<0.001
Operation time (min, median, IQR)	51.0 (35.0–71.0)	74.5 (50.7–91)	<0.001
Total laser energy (J, median, IQR)	1420 (560–3560)	3690 (1902.5–8947.5)	<0.001
<Intra-operative complication; (no, %)>			
Damage of flexible ureteroscopy	2 (0.8)	1 (1.3)	0.578
Ureteral injury Grade 0 Grade 1 Grade 2 Grade 3 Grade 4	225 (98.2) 4 (1.7) 0 (0) 0 (0) 0 (0)	68 (89.4) 5 (6.5) 3 (3.9) 0 (0) 0 (0)	0.001
<Post-operative complication; (no, %)>			
Fever (>38 °C) Phone-call Emergency visit Re-admission	13 (5.6) 0 (0) 1 (0.4) 0 (0)	5 (6.5) 0 (0) 0 (0) 0 (0)	0.573
Clavien–Dindo grade (no, %)			
Grade I Grade II Grade IIIa Grade IIIb Grade IVa Grade IVb Grade V	0 (0) 13(5.6) 1 (0.4) 0 (0) 0 (0) 0 (0) 0 (0)	0 (0) 5 (6.5) 0 (0) 0 (0) 0 (0) 0 (0) 0 (0)	0.573

IQR; interquartile range, * Mann–Whitney U test and chi-squared test.

**Table 3 jcm-12-00301-t003:** Predictive risk factors of stone-free status after retrograde intrarenal surgery for lower pole calices.

<Predictive Factors>	Multivariate Analysis (Odds Ratio, 95% CI, *p*-Value) **
Total stone size (mm, median, IQR)	1.001, 1.000–1.001. *p* = 0.137
Total stone volume (mm^3^, median, IQR)	1.056, 1.015–1.099. *p* = 0.007
CT-value (HU, median, IQR)	1.001, 1.000–1.002. *p* = 0.073
IPA (degree, median, IQR)	0.970, 0.956–0.993. *p* = 0.009
IW (cm, median, IQR)	0.295, 0.121–0.718. *p* = 0.007
IL (cm, median, IQR)	1.119, 0.520–2.276. *p* = 0.756
CPH (cm, median, IQR)	1.207, 0.596–2.441. *p* = 0.601

** Logistic regression analysis. IQR; interquartile range, IPA; infundibulopelvic angle, IW; infundibular width, IL; infundibular length, CPH; caliceal pelvic height.

## Data Availability

This study did not report any data to public data link.

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
