# Peer review of "Single-Session Impact of High-Power Laser with Moses Technology for Lower Pole Stones in Retrograde Intrarenal Surgery: Retrospective Study"

_jcm, 2022, doi:10.3390/jcm12010301_

Round 1

Reviewer 1 Report

Corrections for lower pole "stone". Lack of the word "stone" in the whole manuscript.

Attached my sample of correction.

Author Response

Reply to the Reviewer: 1

Comments to the Author; “Corrections for lower pole “stone”. Lack of the word “stone” in the whole manuscript.”

Reply to reviewer 1; Thank you for your detailed reading my manuscript. According to your pointing-out, I changed the word from “lower pole calices” to “lower pole stones” in whole manuscript. Thank you again. I appreciate your pointing-out. Please check the revised manuscript.

We also had done the proofreading of our English manuscript. I attached the certificate for proofeading. 

Author Response

  • Reply to the Reviewer: 2

・Reviewer comment 1. ; “This makes readers confused. Apparently the paper is a original study but the title says it is a review.”

Reply to reviewer 2; Thank you for your useful comments. I agree with your opinion. According to your suggestion, I revised our manuscript title like below.

“Retrospective review” ⇒ “ Retrospective study”

・Reviewer comment 2. ; “In abstract, retrograde intrarenal surgery needs abbreviation.”

Reply to reviewer 2; Thank you for your detailed awareness. I added the “RIRS” as abbreviation of “retrograde intrarenal surgery” in abstract.”

・Reviewer comment 3. ; “In conclusion of abstract, “Stone volume, IPA, and IW influenced the predictive risk factors related with postoperative SF”.”

Reply to reviewer 3; Thank you for your comment. According reviewer’s comment, I changed expression of this sentence like below.

In abstract and conclusion;

“Stone volume, IPA, and IW influenced the predictive risk factors related with postoperative SF” ⇒ “Larger stone volume, acute IPA, and narrow IW were negative predictors related with postoperative SF status” 

・Reviewer comment 4. ; “Actually, this is not what we deal with lower calyx stones. they are challenging because the anatomy of lower calyx prevents us to reach the stone. Also the fragment clearance after RIRS may be difficult because of the anatomy. So the hypothesis of the study is not supported by a reasonable outline.”

Reply to reviewer 4; Thank you for your comment. Yes, I agree with your thoughts. As you say, ever if we target the lower pole stone, we actually displace the stone or stone fragments from lower pole to upper pole or renal pelvis to easily break the stones. Therefore, we actually target the much tiny stone dust left in lower pole in this study. So, we included those whole circumstances as you say, and evaluated it as lower pole stones.

・Reviewer comment 5. ; “Not scopy but scope. Scopy is the name of technique, the operation. Scope is the tool what we use for performing ureteroscopy.”

Reply to reviewer 5; Thank you for your great teaching for us. I revised it according to your useful suggestion.

In Material and Methods

“ureteroscopy” ⇒ “ureteroscope”

・Reviewer comment 6,7. ; “You mean lower pole stones.”

Reply to reviewer 6,7; Thank you for your pointing-out. I changed it from calices to stones according to your suggestion.

In Material and Methods

“lower pole calices” ⇒ “lower pole stones”

In Material and Methods;

The following sentence; “0, no ureteral lesions or only petechias; 1, mucosal erosion or a mucosal flap without smooth muscle injury; 2, damage to the mucosa and smooth muscle but no adventitia, without a visible retroperitoneal tissue; 3, injury indicating ureteral perforation, involving the full thickness of the ureteral wall, including the adventitia; and 4, total ureteral avulsion.” was omitted from material and Methods.

・Reviewer comment 9. ; “Similarly, if you used a measurement method which defined and introduced already, no need for further explonations. Just cite the original paper.”

Reply to reviewer 9; Thank you for your pointing-out. I omitted this sentence according to your suggestion.

In Material and Methods;

The following sentence “The IPA was measured as the inner angle formed at the intersection of the ureteropelvic (defined by Elbahnasy et al.) and central axes of the lower pole infundibulum. The IW was measured at the narrowest point along the infundibular axis. The IL was measured from the most distal point at the bottom of the infundibulum to a midpoint at the lower lip of the renal pelvis. The CPH was measured as the length of a horizontal line stretching from the lowest point of the calyx to the highest point of the lower lip of the renal pelvis.” was omitted.

・Reviewer comment 10. ; “Yes you used a high power laser but no high power laser settings. Even you have done so, it is still not possible to attribute all the outcomes to the laser. You do not have a comparative study design.”

Reply to reviewer 10; Thank you for your deep comment. Yes, I agree with you. As you said it, only laser is not everything to archive the stone surgery in RIRS. On one point, lower power setting with MT was used, on another point, high Hz(80Hz) with lower energy in MT was used to break the stones. In different circumstances during RIRS, stone basketing was used to archive stone free as much as possible. Therefore, only laser technology is not still everything as you say. In this study, we just evaluated our experiences in stone high volume center as retrospective study. We now have planned to have comparative study.

・Reviewer comment 11,12. ; “In Statistical analysis, wrong words were pointed out; “nominal” and “multivariate”.”

Reply to reviewer 11,12,13; Thank you for your pointing it out. I revised the words according to your suggestion. Thank you again/

“nominal” ⇒ “categorical”

“multivariate” ⇒ “multivariable”

“CT-value” ⇒ “stone density”

・Reviewer comment 14. ; “The exact clinical benefit of the high-power lasers were aimed at this kind of hard stones. Why did not you used higher energy settings to archive stone-free status?”

Reply to reviewer 14; Thank you for your comment. As we described the surgical methods when using Ho-laser, we started the stone breaking of all large or hard stones from high energy with low Hz with MT. After creating stone fragments, we displaced all fragments in one renal calyx. And then, we used the pop-dusting with MT. Therefore, in SF and non-SF we started from high energy setting.

・Reviewer comment 15. ; “What was the point for analysing this factors? If the lower calyx is narrow-angled, narrow itself, of long we should not use high power lasers of something..”

Reply to reviewer 15; Thank you for your great comment. As you said, I also think that high-power laser setting like 0.5*80Hz should not used in those cases. Although we, of course, used the high-power laser system with MT, we didn’t use high-power laser setting with MT in all steps of stone breaking. If we encountered these complicated cases you said, we might use the high energy*lower Hz with MT contact mode to create the moderate stone fragments. After that, we displace it.

・Reviewer comment 16,17,18. ; “In Table2, “No” to “n”, and what does it mean? “less ureteral sheath.

Reply to reviewer 16,17,18; Thank you for your pointing it out. We revised these according to your comments.

In Table 2;

“No,%” ⇒ “no,%”

“Less ureteral sheath” ⇒ “ No use of ureteral sheath”

・Reviewer comment 19. ; “That is really interesting. Using high power laser for having favourable outcomes had ended up with more cumulative energy consumption in non stone-free patients.

Reply to reviewer 19; Thank you for your comment. Yes, this is real clinical practice. As you said, only laser is not everything in stone surgery. Thank you for you’re a lot of insights. I appreciate your useful comments.

Reviewer 3 Report

Paper Review: Single-session impact of high-power laser with Moses technology for lower pole calices in retrograde intrarenal surgery: Retrospective review

The authors evaluated the efficacy of high-power holmium lasers with Moses technology for treatment of lower pole stones. The cohort was subdivided into stone free and non-stone free on the first post-operative day and after 1 month.

The authors report a stone free rate of 56.7% on POD 1 and 75.1% at 1 month following the procedure.

In the non-stone free cohort, the reported larger stone size and volume, as well as a higher stone density. Unfavorable anatomy was also found to be associated with a lower stone free rate.

The authors conclude that high-power holmium lasers with Moses technology for treatment of lower pole stones is “a valuable option in achieving a stone free status and patient safety”.

It is my understanding that the study has actually shown that the stone free rate when treating smaller and softer stones is higher than when treating harder and larger stones. This has been reported in multiple studies and is not novel. The fact the utilizing Moses technology doesn’t improve the stone free rate is interesting and worth reporting on, but this should be done comparing Moses to non-Moses lasers.

Author Response

  • Reply to the Reviewer: 3

Comments to the Author; “The authors evaluated the efficacy of high-power holmium lasers with Moses technology for treatment of lower pole stones. The cohort was subdivided into stone free and non-stone free on the first post-operative day and after 1 month. The author report a stone free rate of 56.7% on POD1 and 75.2% at 1-month following the procedure. In the non-stone free cohort, the reported larger stone size and volume, as well as a higher stone density. Unfavorable anatomy was also found to be associated with a lower stone free rate. The authors conclude that high-power holmium lasers with Moses technology for treatment of lower pole stones is a valuable option in archiving a stone free status and patient safety. It is my understanding that the study has actually shown that the stone free rate when treating smaller and softer stones is higher than when harder and larger stones. This has been reported in multiple studies and is not novel. The fact the utilizing Moses technology doesn’t improve the stone free rate is interesting and worth reporting on, but this should be done comparing Moses to non-Moses lasers.”

Reply to reviewer 3; Thank you for your evaluation for our study. Yes, I agree with your opinion. It was surprising for us that using Moses technology didn’t improve the stone free status for moderate lower pole stones compared to historical reports with regular laser mode. This is a real outcome in our high-volume stone center. However, we have a impression such as being faster operative time than before. Of course, as you say, we will have to compare Moses to non-Moses laser. This study was focused on just outcome of Moses technology use in lower pole stones to want to know whether Moses technology improves the stone free or not compared to previous reports because there are few reports regarding with Moses technology for lower pole stones. Of course, we have evaluated now the comparison study.

Round 2

Reviewer 3 Report

The manuscript does not present any novel data. Although the authors report on results of utilization of Moses technology in the treatment of lower pole stones, the fact that the stone free rate is lower in treating larger and harder stones has been extensively reported and is not new. I believe a comparison between Moses and non- Moses would be interesting to report on. 

Author Response

  •  Response to Reviewers
  • Reply to the Reviewer: 3

Comments to the Author; “The manuscript does not present any novel data. Although the authors report on results of utilization of Moses technology in the treatment of lower pole stones, the fact that the stone free rate is lower in treating larger and harder stones has been extensively reported and is not new. I believe a comparison between Moses and non- Moses would be interesting to report on.”

Reply to reviewer 3; Thank you for your comment and suggestion again. I know and agree with what you say that the MOSES should compare to non-MOSES in lower pole stones. However, this our article just compared with SF and non-SF in MOSES mode for lower pole stones against your suggestion. As we mentioned in first reply for you, now we prospectively are going on accumulating the data in non-MOSES and MOSES for all kidney stones. Therefore, it is difficult to immediately assess the comparison which you want. Sorry for our inconvenience. However, there are few reports regarding with the result of RIRS with Moses mode in lower pole stones. Therefore, we think that this retrospective study might be also one of meaningful data in clinical practice. Of course, we will access the comparison between MOSEES and non-MOESE after prospective study. We want you to wait our next trial. Again, it is sorry for our inconvenience that we can’t immediately reply your whole suggestion.
